# Additive Manufacturing of Honeycomb Lattice Structure—From Theoretical Models to Polymer and Metal Products

**DOI:** 10.3390/ma15051838

**Published:** 2022-03-01

**Authors:** Tomáš Goldmann, Wei-Chin Huang, Sylwia Rzepa, Jan Džugan, Radek Sedláček, Matej Daniel

**Affiliations:** 1Department of Mechanics, Biomechanics and Mechatronics, Faculty of Mechanical Engineering, Czech Technical University in Prague, Technická 1902/4, 16000 Prague, Czech Republic; tomas.goldmann@fs.cvut.cz (T.G.); radek.sedlacek@fs.cvut.cz (R.S.); matej.daniel@fs.cvut.cz (M.D.); 2Laser and Additive Manufacturing Technology Center, Industrial Technology Research Institute, No. 8, Gongyan Rd., Liujia Dist., Tainan City 734, Taiwan; 3Mechanical Testing and Thermophysical Measurement Department, COMTES FHT a.s., Průmyslová 995, 33441 Dobrany, Czech Republic; sylwia.rzepa@comtesfht.cz (S.R.); jan.dzugan@comtesfht.cz (J.D.)

**Keywords:** lattice structures, titanium alloys, laser powder bed fusion, additive manufacturing, numerical simulation, analytical modeling

## Abstract

The study aims to compare mechanical properties of polymer and metal honeycomb lattice structures between a computational model and an experiment. Specimens with regular honeycomb lattice structures made of Stratasys Vero PureWhite polymer were produced using PolyJet technology while identical specimens from stainless steel 316L and titanium alloy Ti6Al4V were produced by laser powder bed fusion. These structures were tested in tension at quasi-static rates of strain, and their effective Young’s modulus was determined. Analytical models and finite element models were used to predict effective Young’s modulus of the honeycomb structure from the properties of bulk materials. It was shown, that the stiffness of metal honeycomb lattice structure produced by laser powder bed fusion could be predicted with high accuracy by the finite element model. Analytical models slightly overestimate global stiffness but may be used as the first approximation. However, in the case of polymer material, both analytical and FEM modeling significantly overestimate material stiffness. The results indicate that computer modeling could be used with high accuracy to predict the mechanical properties of lattice structures produced from metal powder by laser melting.

## 1. Introduction

Functional properties of products manufactured by classic subtractive methods can be predicted with high accuracy as there is extensive engineering knowledge on the mechanical behavior of materials and the effect of their processing. However, additive manufacturing (AM) builds the produced layer after layering and this process influences mechanical behavior to a large and still partially unknown extent. There is a wide range of AM methods currently forming the product from filaments, droplets, or powder of raw material made of polymers, metals, ceramics, or even a combination of these materials [1]. The imperfections, which can occur during such complex AM processes, namely pores, part distortion, or lack of fusion defects, should not also be neglected in the design process [2] but they also cannot be predicted explicitly. It has been shown in numerous studies that the properties of AM products largely depend on the orientation of the structure during the manufacturing process, size of the structure, or set of process parameters [3]. In addition, each AM method has its specifics, which makes it difficult to transfer knowledge between them [4]. There is an ongoing effort to provide methods and data that will help to understand the limitations of individual AM methods and allow the safe design of AM products.

One of the advantages of AM is the ability to produce unique lattice structures with tunable mechanical, acoustic, thermal, or electrical properties that can hardly be manufactured otherwise [5]. New design possibilities also bring new problems in modern engineering design that largely relies on computer modeling. The transition from a computer model to a fully functional 3D component is not in many cases straightforward, especially for complex geometries such as lattice structures. The lattice structure consists of a network of relatively thin interconnected walls or struts. Such structures are characterized by a high surface area to volume ratio that makes them prone to surface imperfections during the building process [6].

It is the aim of this study to provide a direct comparison between the computer model of honeycomb lattice structure and its mechanical properties. To test the accuracy of theoretical predictions, identical samples were consequently produced using PolyJet technology from Vero PureWhite polymer and two independent laser powder bed fusion systems from stainless steel 316L and Ti6Al4V titanium alloy. Previous studies dealing with honeycomb structure tested the lattice by compression [7,8,9], bending [10], or rely on theoretical analysis only [11]. Within the presented study, the mechanical properties of the honeycomb structure were predicted using the analytical and the numerical model. The accuracy of predictions was verified by experimental results using three different materials in the tensile test. The tensile test is more appropriate for additively manufactured materials as it can identify defects of materials due to lack of fusion, keyhole collapse, or gas porosity more effectively than compression tests [7]. In a compression test, material defects may close up and do not have a significant impact on behavior under compression modes of loading [12].

## 2. Materials and Methods

### 2.1. Honeycomb Lattice Specimen

A hexagonal lattice was chosen as a representative structure in this study. Hexagonal lattice has the morphology of bee honeycombs and is commonly used as infills of AM parts to keep parts lightweight [5]. This structure is also a schoolbook example for modeling properties of cellular solids [13].

A standard tensile specimen according to ISO 527 was modified by adding a central part consisting of a uniform honeycomb structure (Figure 1). The honeycomb structure consists of repeating unit cells of hexagonal shape. Regular hexagonal cells consist of six same struts of inner length $l_{b}$, while the outer angles are 120°. The honeycomb cells have the same thickness *t* of the cell walls throughout the entire pattern. The node is defined as an element of the lattice structure where individual struts are connected. The unit cell has a wall thickness of 1 mm and inner length of 3 mm. The thickness of the tensile samples was designed to be 4 mm. Design software SolidWorks (version 2019 SP5, Dassault Systems SolidWorks Corp., Waltham MA) was used to create geometry models of the tensile specimens (Figure 1).

The effective density of the proposed structure $\rho^{*}$ could be estimated as follows [14]:(1)$$\frac{\rho^{*}}{\rho_{s}}=1-\frac{l_{b}^{2}\left(1+\sin\theta \right)}{l^{2}\left(1+\sin\theta \right)}$$
where $\rho_{s}$ is the density of the material, and $l_{b}$ and *l* are the inner and the outer length of the unit cell and $\theta =30^{{}^{\circ}}$. The outer length of the unit cell can be determined as follows.
(2)$$l=l_{b}+\frac{t}{2}\frac{1}{\cos\theta }$$

### 2.2. Analytical Model

The analytical model is based on an analysis of Gibson and Ashby (1997) that was further upgraded by Malek and Gibson (2015). The stiffness of the honeycomb structure in tension is expressed in terms of effective modulus $E^{*}$. The effective modulus is a modulus of hypothetical homogeneous material that has the same stress–strain behavior as the porous lattice. In addition to bending, a shear is calculated according to the Timoshenko beam theory while deformations within the node are neglected. Considering deflection due to tension, shear, and bending, the effective stiffness of the regular hexagonal unit cell could be expressed as follows Figure 2 [14]:(3)$$\frac{E^{*}}{E_{s}}={\left(\frac{t}{l_{b}}\right)}^{3}\left(\frac{1}{1+\left(5.4+1.5\nu \right){\left(\frac{t}{l_{b}}\right)}^{2}}\right)$$
where $E_{s}$ and ν are the Young’s modulus of the material and its Poisson’s ratio. However, Equation (Equation 3) was derived for the unit cell of an infinite honeycomb structure.

To calculate the stiffness of the experimental specimen, we should consider a finite number of the cells and the fact that cells at the boundaries do not share a wall with neighbors. It holds also for the first and the last rows of cells along the longitudinal axis of the specimen. The Young’s modulus of individual cell row can be expressed as follows:(4)$$E_{i}=\frac{nE_{1}+E_{2}}{n+k}$$
where $E_{1}$ is Young’s modulus of the inner cell calculated from Equation (Equation 3) and $E_{2}$ is Young’s modulus of the boundary cell calculated from Equation (Equation 3) by taking a twice as large value for *t*. For odd rows (i=o), i.e., rows with 5 honeycomb cell in Figure 1, n=4 and k=1 while for even rows (i=e), i.e., rows with 4 honeycomb cells in Figure 1, n=3 and k=2. Parameter *k* accounts for empty cells in even rows. The Young’s modulus of the entire structure could be expressed as follows:(5)$$E^{*}=\frac{E_{e}E_{o}\left(n_{e}+n_{o}\right)}{n_{e}E_{o}+n_{o}E_{e}}$$
where $n_{e},E_{e}$, and $n_{o},E_{o}$ are the number of even and odd rows and their Young’s moduli, respectively. For tspecimen used in this study (Figure 1), it is is $n_{e}=n_{o}=5$ as only half of the first and the last odd row is included in the lattice structure.

### 2.3. Numerical Model

Finite element analysis (FEA) models of the honeycomb structure were constructed with ABAQUS/CAE (version 2019.HF9 Dassault Systemes Simulia Corp., Johnston, RI, USA). FEA analysis assumes the same material parameters as the analytical model. As the analytical model assumes only linear elasticity, the FEA model adopts a linear elastic model as well. Studied materials are, therefore, defined by two material constants provided in Table 1.

The methodology of the simulation follows the study of Alwattar and Mian, 2019 [15]. The honeycomb lattice (Figure 3a) was studied along with an equivalent elastic material model (Figure 3b). The honeycomb lattice structure and the equivalent elastic block have the same external dimensions. The equivalent material model is a solid material model that contains no struts but has a mechanical response equivalent to a honeycomb lattice structure, i.e., the same stress/strain dependence. The equivalent elastic mechanical property considered within this study is the effective Young’s modulus $E^{*}$.

Both models have been meshed in ABAQUS CAE with quadratic 3D deformable elements of brick shape with 20 nodes—C3D20. Mesh size was selected according to convergence error analysis. To describe the behavior of the entire honeycomb lattice specimen (Figure 1), it is important to select boundary conditions equivalent to the experiment. For tensile test simulation, the model (Figure 3a) was clamped between two plates. The displacement of the specimen is prescribed to remain zero at the lower plate (encastre boundary condition), while loading is prescribed via the displacement in the longitudinal direction at the upper plate. The load was chosen so that the stress within the model will not exceed the yield stress of the material (Table 1).

### 2.4. Additive Manufacturing of Specimens

The specimens were fabricated from three various materials: one polymer material and two metal alloys. Polymer specimens in this study were fabricated using 3D printer Stratasys J750${}^{\mathrm{TM}}$ (Stratasys Inc., Eden Prairie, MN, USA), while metal specimens were manufactured from titanium alloy Ti6Al4V and stainless steel 316L. Two main reasons influenced material choice: first, the chosen materials are commonly used in current AM products. Secondly, the AM technology of these materials is well known at the workspace of the authors [16,17,18] and they cover relative heterogeneity in materials used for additive manufacturing, assuring observable differences.

Samples were fabricated from Vero PureWhite RGD837 (acrylic-based photopolymer) with support material SUP706. SUP706 forms the support structure during the PolyJet AM process. The thickness of the layer was 27 μm with a resolution of 600 dpi at setting for high-speed printing. The support material SUP706 was removed chemically.

Laser powder bed fusion (LPBF) is currently the most prominent method that allows production from a wide range of metal materials. Within the proposed study, two independent laser melting printers were used. The samples made of titanium alloy Ti6Al4V were produced using the 3D printer M2 cusing (Concept Laser GmbH, Lichtenfels, Germany) from Concept Laser CL 42Ti Grade 2 powder consisting of particles with sizes ranging from 20 to 50 μm. The process parameters were the following: The build chamber was not pre-heated; the laser beam power was 200 W; the scan speed was 0.5 m/s; the layer thickness was 20 μm; and the offset distance was 75 μm. Concept Laser’s ‘island’ scan strategy was applied. Heat treatment was performed to relieve residual stresses under an argon atmosphere. The samples were heated up in 4 h to 840 °C and the temperature was maintained for 2 h.

The second set of metallic samples was manufactured using an Industrial Technology Research Institute (ITRI) LPBF deposition system AM250. Stainless steel 316L powder of average particle size of 35 μm, with sizes ranging from 20 to 53 μm was used. A laser power of 160 W, the scan speed of 1100 mm/s, hatching spacing of 0.1 mm, and layer thickness of 30 μm was used. The part building process takes place inside an enclosed chamber filled with nitrogen gas to minimize the oxidation of powdered material. Solution heat treatment was performed at 950 °C for 2 h.

### 2.5. Experimental Testing

The dimensions of the honeycomb structure were evaluated using 3D scanner ATOS by GOM (Capture 3D, Inc., Santa Ana, USA) providing high-precision three-dimensional data. The system is equipped with two high-resolution cameras of 12 MPx with a measurement volume of 320 × 240 × 320 mm${}^{3}$ and an accuracy of 5 μm. All specimens were measured before experiments.

An electromechanical testing system, Mayes, was used to conduct the uniaxial tests, and the Aramis SRX system with a 4-megapixel camera was utilized to monitor 2D strain fields with high resolution. All tests were performed at a constant strain rate of 0.04 s at room temperature. The difference in effective modulus between analytical prediction and experimental data was evaluated by z-test (Python 3.9.5, library statsmodels [23]).

## 3. Results

Polyjet and laser powder bed fusion can produce honeycomb lattice structures with high accuracy (Table 2). The relative density of the produced structure, i.e., the density with respect to the density of bulk material, is 19.2 (Equation (Equation 1)). The experimental tensile test proved good repeatability between individual samples for all materials (Figure 4).

The lowest stiffness was observed for Stratasys Vero PureWhite (Figure 4), as could be expected from the material properties of the bulk polymer (Table 1). Titanium alloy Ti6Al4V exhibits limited extensibility, and the elongation at break is on average 3.5 ± 0.3% (Figure 4 and Figure 5c). Stainless steels honeycomb specimens are characterized by extensive plastic deformation where the elongation at break reaches 25.2 ± 0.2% (Figure 5d). Specimens made of stainless steel 316L exhibited progressive failure behavior manifested by multiple stress peaks recorded in the stress–strain curve (Figure 6). At a certain force value, a fracture of a single strut was observed and further loading resulted in the fracture occurrence in other struts until complete specimen failure.

The von Mises stress distribution for polymer specimen is shown in Figure 5a. The maximum stress is observed at the nodes of individual honeycomb cells. These are also the points where the material cracks after loading (Figure 5b,c). The analysis of fracture surfaces of metal specimens after the tensile test was performed using scanning electron microscope JEOL IT 500 HR (JEOL Ltd., Tokyo, Japan) working under the secondary (SED) and backscattered electrons (BED-C) regime. In the case of both materials, titanium alloy Ti6Al4V and stainless steel 316L, fracture surfaces have a ductile character with dimple morphology, as observed in Figure 7. The surfaces revealed an appearance of transcrystalline fracture. In addition, pores and lack-of-fusion defects were observed.

In order to compare the results between the model and the experiments, the values of relative effective modulus $E^{*}/E_{s}$ are expressed in Table 3. Experimentally determined effective Young’s modulus is significantly lower than values predicted by the analytical model (z-test z=−144.2, p<0.001 for Stratasys Vero PureWhite, z=−4.7, p<0.001 for Ti6Al4V, and z=−2.37, *p* = 0.018 for 316L specimens). However, the effective Young’s moduli obtained for Ti6Al4V and 316L specimens are close to the value predicted by the analytical model. Finite element simulation exhibits an almost perfect match with experimental data for titanium alloy and stainless steel (z-test z=0.02, p=0.98 for Ti6Al4V, and z=0.003, p=0.99 for 316L specimens). We observed a considerable discrepancy between the predicted and the measured stiffness of lattice structure for the Stratasys Vero PureWhite material (z=−122.16, p<0.001).

## 4. Discussion

The current study demonstrates that the theoretical predictions of structure stiffness are in good agreement with the experimental data for metal materials prepared by LPBF but considerably overestimate properties of the same lattice structure manufactured by the PolyJet method.

Our results indicate that the FEA model is more accurate than an analytical model of the honeycomb lattice structure. Contrary to our results, Malek and Gibson, 2015 [14], reported an excellent correlation between the analytical and the FEA model. However, they studied only a single cell without considering the effect of the boundaries. The data reported by Gibson, 1981 [24], and mentioned in the work of Malek and Gibson, 2015, indicate that the theoretical model might slightly overestimate structural stiffness. However, this difference is small in the order of 0.1%. From our results, the difference between the analytical and the numerical models is approximately 0.8% in the relative effective modulus (Table 3). It could be explained by neglecting the deformation in the nodes by the analytical model. FEA simulation presented in Figure 5 indicates that the largest stress is acting in nodes that affect their deformation. Failure in nodes was also mostly observed during experiments.

Within this study, PolyJet technology was chosen as a representative of modern polymer manufacturing. PolyJet technology was selected as it has limited sensitivity to the direction of printing [25] in comparison to fused filament fabrication [26]. However, the lattice produced by PolyJet provided the worst results in terms of the predictivity of mechanical properties. Within the current study, the mechanical properties of materials obtained using identical additive manufacturing are assumed as input parameters into theoretical models (Table 1). However, studies mentioned in Table 1 use standard tensile test specimens that are much larger than individual elements of the hexagonal lattice structure. Furthermore, the properties of the polymer material are not fixed and depend on production settings, i.e., the layer thickness or the specimen building orientation [27] and likely the size of the specimen. The Stratasys reports Young’s modulus ranging from 1000 MPa to 2000 MPa. Birosz et al., 2021 [28] even reported the approximate modulus of material Vero PureWhite to be 766 MPa. Good repeatability of experiments (Figure 4) indicates that there is a global systematic deterioration in mechanical properties. We, therefore, could conclude that the most probable cause of the difference is the difference between the mechanical properties of structural members of a lattice and the properties of the bulk polymer. When printing small struts of lattice structure using PolyJet technology, there is potential mixing of the supporting and the build material at a surface. During post-processing, the support material is removed. As a surface-to-volume ratio is high, it may induce considerable defects to the surface of polymer material that lowers the Young’s modulus. A similar effect could be observed in metal specimens with high porosity [29]. Polymer material might also exhibit viscoelastic behavior for cycling load or in long-term static load. Moreover, these effects were not observed in the current study, and they should be considered in the design of polymer lattice structure [30]. The effect of the size of the structural elements manufactured by PolyJet technology on the mechanical properties warrants further investigation.

It should be noted that the results of the current study are valid for a lattice structure of given geometry and size. For metallic materials, Pehlivan et al., 2020 [31] showed that the mechanical properties of individual struts depend on their size and building direction during manufacturing. This has been further confirmed in the study of Mertova et al., 2020 [3] where surface area was identified as a decisive factor contributing to mechanical performance degradation. Lattice structures are characterized by a large surface-to-volume ratio that improves their thermal dissipation efficiency [32] but could result in material defects due to the surface’s lack of fusion [6]. There are also possible internal defects such as the lack of fusion, keyhole collapse, or gas porosity due to limited track of the laser beam during the melting process on small cross-sections. As suggested by Pehlivan et al., 2020 there exists a critical size of struts, after which the effect of orientation and size on mechanical properties are irrelevant. Our simulations and experiments confirm this conclusion and show that, for a large lattice structure, the properties of individual elements are averaged and a simple homogeneous model predicts its behavior with good accuracy.

## 5. Conclusions

The study is intended to test the applicability of the theoretical prediction of honeycomb lattice structure mechanics loaded in tension. We have shown that the stiffness of the metal honeycomb lattice structure produced by LPBF could be predicted with high accuracy by the FEA model. The linear analytical model slightly overestimates global stiffness but may be used as the first approximation. However, using bulk properties of polymer from PolyJet technology results in the unrealistic high estimation of lattice material stiffness. The results of the study show that numerical modeling could be effectively used in the future design of metal lattice structures.

## Figures and Tables

**Figure 1 materials-15-01838-f001:**
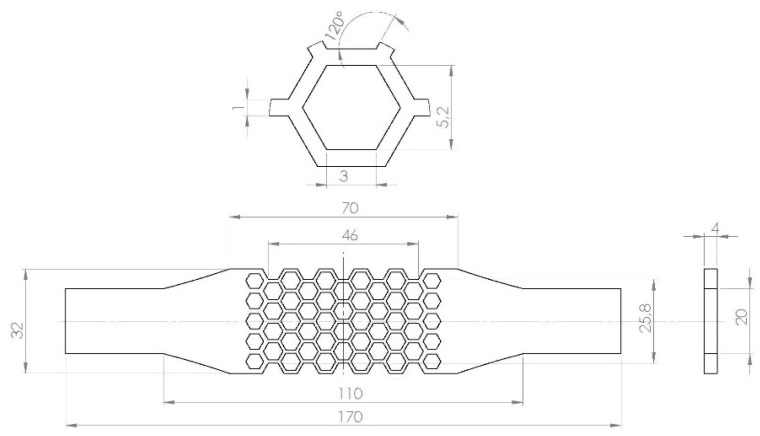
Geometry of tensile specimen with honeycomb structure.

**Figure 2 materials-15-01838-f002:**
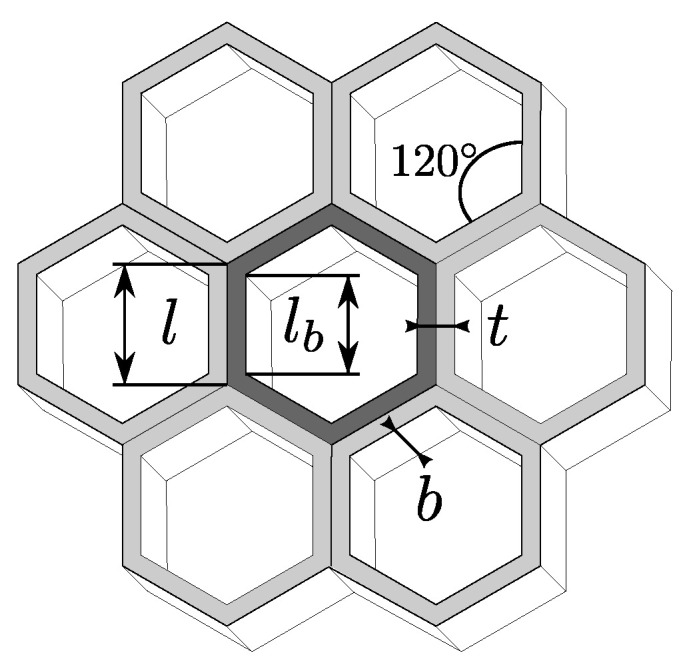
Dimensions of the hexagonal honeycombs [14]. A unit cell of material is highlighted.

**Figure 3 materials-15-01838-f003:**
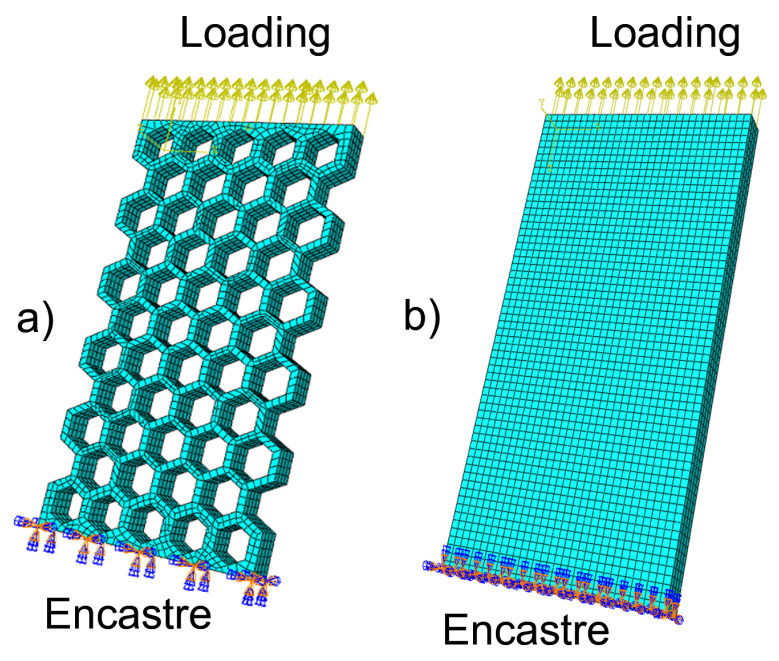
(**a**) Finite element model of the regular honeycomb structure. (**b**) Equivalent homogeneous material with the effective modulus $E^{*}$. The effective modulus $E^{*}$ is chosen so that the equivalent material model has the same mechanical response as the honeycomb lattice structure.

**Figure 4 materials-15-01838-f004:**
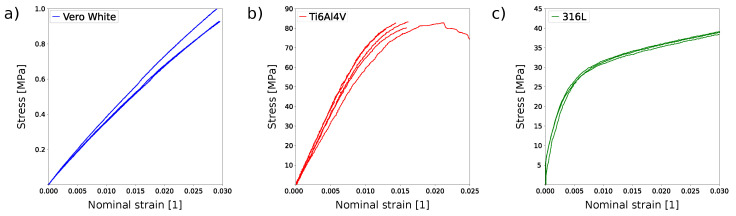
Stress–strain curves for regular honeycomb structure additively manufactured from (**a**) Stratasys Vero White polymer, (**b**) titanium alloy Ti6Al4V, and (**c**) stainless steel 316L. Curves were obtained in tensile tests.

**Figure 5 materials-15-01838-f005:**
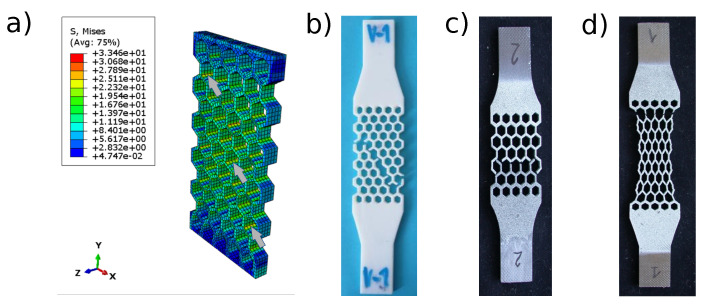
(**a**) von Mises stress distribution for Stratasys Vero PureWhite specimen loaded by the axial tensile force of 200 N. Value of stress is expressed in MPa. Values of stress are expressed in MPa. Maximum stress is identified at connecting nodes of individual honeycomb cells. Few nodes with stress concentration are marked by arrows. Experimental specimens from Stratasys Vero White polymer (**b**), titanium alloy Ti6Al4V (**c**) and stainless steel 316L (**d**) show fractures at connecting modes.

**Figure 6 materials-15-01838-f006:**
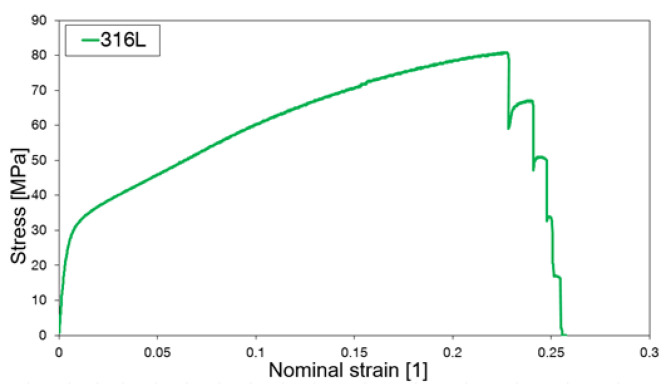
Stress–strain curves for regular honeycomb structure fabricated from stainless steel 316L with visible force peaks corresponding to the failure of individual struts. Curve was obtained in the tensile test.

**Figure 7 materials-15-01838-f007:**
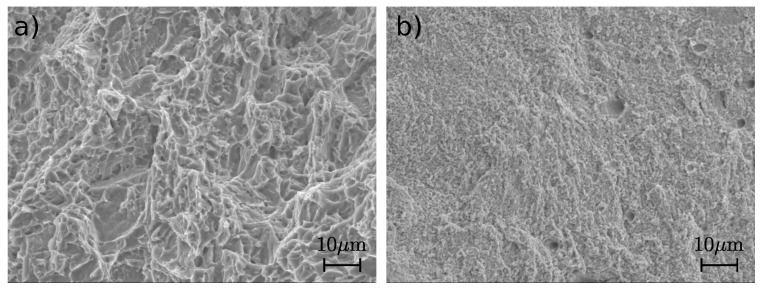
Micrographs of fracture surfaces of the specimens after tensile test made of (**a**) titanium alloy Ti6Al4V and (**b**) stainless steel 316L.

**Table 1 materials-15-01838-t001:** Young’s modulus (*E*) and Poisson’s ratio (ν) of tested materials.

Material	*E* (GPa)	ν (1)	Yield Stress (MPa)	Source
Stratasys Vero PureWhite	1.9	0.33	50	[19,20]
Ti6Al4V	119	0.35	1100	[21]
316L	183	0.30	600	[22]

**Table 2 materials-15-01838-t002:** Dimensions of specimens from additive manufacturing.

Material	AM Machine	Technology	$l_{b}$ (mm)	*t* (mm)	*b* (mm)
Vero PureWhite	Stratasys J750	polyjet	3.02 ± 0.18	1.01 ± 0.05	4.02 ± 0.05
Ti6Al4V	Concept Laser	DLMS	3.06 ± 0.15	1.03 ± 0.02	3.99 ± 0.01
316L	ITRI	DLMS	3.07 ± 0.16	0.98 ± 0.01	3.97 ± 0.01

**Table 3 materials-15-01838-t003:** Effective Young’s modulus in tension. Comparisons between prediction from analytical, numerical model and experimentally determined values for Stratasys Vero PureWhite, Ti6Al4V, and 316L. $E^{*}$ denotes effective Young’s modulus, and $E_{s}$ is Young’s modulus of the bulk material.

	Material	$E^{*}$	$E^{*}/E_{s}$
Analytical model	Vero PureWhite	139.87 MPa	7.36%
	Ti6Al4V	8.73 GPa	7.34%
	316L	13.53 GPa	7.39%
FEA	Vero PureWhite	124.93 MPa	6.57%
	Ti6AL4V	7.28 GPa	6.12%
	316L	11.59 GPa	6.33%
Experiment	Vero PureWhite	33.16 MPa ± 2.57 MPa	1.75 ± 0.06%
	Ti6Al4V	7.26 GPa ± 0.64 GPa	6.11 ± 0.54%
	316L	11.54 GPa ± 1.18 GPa	6.31 ± 0.64%

## Data Availability

The data that support the findings of this study are available from the corresponding author, W.-C.H., upon reasonable request.

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
