# Peer review of "Additive Manufacturing of Honeycomb Lattice Structure—From Theoretical Models to Polymer and Metal Products"

_materials, 2022, doi:10.3390/ma15051838_

Round 1

Reviewer 1 Report

The authors compare mechanical properties of polymer and metal honeycomb lattice structures between computational model and experiments. The manuscript could be of interest to scientific community. Some interesting results are obtained. I suggest that the manuscript be corrected before its possible publication.

  1. A novelty of the research work is missing and also a comparison with earlier work is also required.
  2. In general, the manuscript needs a grammar revision, since several mistakes were observed along it. (ie, Lines 92, 120, 156, 175, 196).
  3. Poisson's ratio is not in equation 3.
  4. Powder bed laser fusion or laser powder bed fusion? Unify in manuscript.
  5. Ensure that the units are consistent in the manuscript.
  6. It should be mentioned why the materials under study were chosen and not others.
  7. In the polymer and titanium specimens, the multiple stress peaks were not observed? Why not?
  8. Regarding the simulation, did you perform a mesh convergence study? It is important to mention it in the manuscript.
  9. How can the fracture of the specimens be considered, ductile or brittle?
  10. Line 196: must be table 3.
  11. What is the reason for the discrepancy in the value of E* for the polymer?
  12. Considered that the polymer is a viscoelastic material? From a mechanical point of view, it should not be directly compared to a metal.

Author Response

Reviewer #1

The authors compare mechanical properties of polymer and metal honeycomb lattice structures between computational model and experiments. The manuscript could be of interest to scientific community. Some interesting results are obtained. I suggest that the manuscript be corrected before its possible publication.

Authors response:

We thank the reviewer for the comments.

A novelty of the research work is missing and also a comparison with earlier work is also required.

Authors response:

            The novelty of the research was discussed in the Discussion section. We agree with the reviewer, that it should be stated explicitly in the Introduction. Therefore a revised manuscript contains statement of novelty in the Introduction.  

Text Changes:

The revised manuscript contains a paragraph discussing the novelty of the current study in Introduction.

In general, the manuscript needs a grammar revision, since several mistakes were observed along it. (ie, Lines 92, 120, 156, 175, 196).

Authors response:

We thank the reviewer for the comment. The manuscript was edited by an English native speaker.

Text Changes:

Grammatical changes through the manuscript, not marked in the revised manuscript.

Poisson's ratio is not in equation 3.

Authors response:

The conversion between the MS Word and OpenOffice Writer used by some co-authors renders contents of the equations unreadable.  Therefore the whole manuscript was transformed into LaTeX to improve readability and formatting. Equations were checked in the revised manuscript.

Text Changes:

All equations were rewritten to LaTeX and checked.

Powder bed laser fusion or laser powder bed fusion? Unify in manuscript.

Authors response:

The correct term is laser powder bed fusion according to ISO/ASTM 52911-1:2019.

Text Changes:

The expression for laser powder bed fusion was unified in the manuscript. Changes are marked in the text.

Ensure that the units are consistent in the manuscript.

Authors response:

The units were checked in the manuscript.

Text Changes:

Corrections of missing units for micrometers in subsection 2.4. All dimensions for additive manufacturing precision are expressed in micrometers in revised manuscript. 

It should be mentioned why the materials under study were chosen and not others.

Authors response:

We agree.

Text Changes:

The revised manuscript contains a paragraph about the selection of materials. Three new citations were added to support the statement.

In the polymer and titanium specimens, the multiple stress peaks were not observed? Why not?

Authors response:

Multiple peaks were observed for all materials at the nodes of the individual cell. We agree, that the nodes are not defined but referred to in the manuscript. The node is defined as an element of the lattice structure where individual struts are connected. Therefore, the revision manuscript was extended to define the unit cell and its elements more precisely.

Text Changes:

section 2.1. contains a definition of unit cells, struts, and nodes

section 3 Results states: The maximum stress is observed at the nodes of individual honeycomb cells.

Fig. 7: few points of peak stress are depicted by arrows

Regarding the simulation, did you perform a mesh convergence study? It is important to mention it in the manuscript.

Authors response:

The mesh size was chosen based on convergence study. The initial estimation of element size was taken from the geometry of the STL file. In further steps, the same loading conditions were simulated with finer meshes. It was shown, that the peak stress and displacements converge.

Text Changes:

A convergence study is mentioned in the revised manuscript, section 2.3.

How can the fracture of the specimens be considered, ductile or brittle?

Authors response:

Additional analysis of fracture surfaces of metal specimens was performed. The fractures have ductile character. 

Text Changes:

The revised manuscript contains an additional paragraph in the Results section showing the results of fracture surface analysis. The figure was added showing the micrographs of fractured surfaces of the specimens after the tensile test for metal specimens.

Line 196: must be table 3.

Authors response:

We agree.

Text Changes:

Corrected in revision manuscript.

What is the reason for the discrepancy in the value of E* for the polymer?

Authors response:

We believe, that the main reason for the discrepancy in the value of effective elastic modulus for the Stratasys VeroWhite polymer is the difference between the assumed Young's module in Eq. 3 and the real Young's module. When printing small struts of a lattice structure, there is potential mixing of the supporting and the build material at surface layers. During post-processing, the support material is removed. As a surface-to-volume ratio is high, it may induce considerable defects to the surface of polymer material that lower Young's modulus. A similar effect could be observed in metal specimens with high porosity. To prove this hypothesis, we are currently conducting a series of experiments with variable geometry of polymer tensile specimens combined with Raman spectroscopy to detect traces of support material within the AM specimens.     

Text Changes:

The discussion section is expanded to provide a hypothesis of deteriorated mechanical properties for small samples produced using PolyJet technology. Additional reference Choren et al., 2013 was added that shows the effect of porosity on Young's modulus.

Considered that the polymer is a viscoelastic material? From a mechanical point of view, it should not be directly compared to metal.

Authors response:

We thank the reviewer for pointing this out. We agree that the polymer is an elasto-visco-plastic material. However, within the measurements, we have not observed any signs of viscoelasticity like creep or stress relaxation. Therefore, we may consider the material as elastic for low forces and load rate used in the experiment.  

Text Changes:

The paragraph was added to the Discussion section dealing with the effect of viscoelasticity in lattice structures. A recent reference on the study of viscoelasticity in the three-dimensional printed polymeric lattice structures was a

Reviewer 2 Report

Dear Authors,

I have read manuscript titled: “Additive manufacturing of honeycomb lattice structure – from theoretical models to polymer and metal products” with great attention.

The paper provides valuable experimental investigations, but I have major concerns about experimental tests, used specimens and the results which were obtained.

The issues to be considered by the Authors:

Line 72-73: The sentence “The design cell…” is incomprehensible and requires clarification.

Line 74-75: Authors declare that the computer models were created using SolidWorks software although section 2.3 describing numerical modeling depicts FE software ABAQUS. It is not clear what for SolidWorks software was used.

Line 82: It is recommended to denote the indicated hexagonal cell angle of 30 degrees in the Figure 2 together with other parameters.

Line 82: parameter l previously was named as outer length, it is recommended to stay with the uniform terminology throughout the text.

Line 97: statement depicting the terms of the equation 3 includes Poisson’s ratio although the equation does not embed this parameter. The role of the  Poisson’s ratio has to be clarified.

Section 2.3: This section describes the numerical modeling however, the description of the material models requires clarification, because it is not clear what type of material model was used. Does authors use the same material models for polymer and metallic materials? The reference to the table 1which depicts elastic modulus and Poisson’s ratio is not enough adequately describe the material model. Without yielding and/or strength limits it is not clear how the analysis is terminated, this has to be clarified. What is more the description of loading and boundary conditions is missing.

Figure 3: The title of the figure states that the depicted models in the figure have the same stress-strain response. This statement requires augmentation, graphical representation is recommended.

Line 133: The statement “Polymer specimens in this study were fabricated using Stratasys J750” requires clarification. What is meant by Stratasys J750? If it’s 3D printer it has to be identified in the text.

Line 135: The purpose of the support material SUP706 has to be clarified.

Line 141: What is meant by “cusing machine”?

Line 158: It has to be identified what kind of equipment is entitled as “ATOS by GOM”.

Figure 4: It has to be clarified how the results presented in the figure were obtained. Is it experimental results, numerical or comparison of both.

Figure 5: The previous comment is applicable also for the Fig. 5.

Line 180 and Line 182: authors provide average elongation in percentage and refer to the Figure 4 although the strain in fig. 4 is not in percentage. The reference is inexpedient. Either the strain in Fig. 4 or in the text has to be changed to resemble each other.

Line 191-192: The statement “The maximum stress is observed at the nodes” requires clarification as the nodes mentioned by the authors are not identified. It is recommended to consider the identification of the considering nodes in Fig. 6.

Line 193: The statement “Similar results were observed…” has to be supported by the figure.

Line 231: Authors state that the difference of the results is 1% although from the table 3 it is closer to 2%.

Line 232-233: The statement “FEA simulation presented in Fig. 6…” has to be supplemented, the Fig. 6 has clearly depict the nodes of stress concentration.

Some general comments:

From the context the experimental program is not clear. Were the tests performed by the authors or the other researches. Were the material properties tested in the lab by the authors or just the results available from the other research used. It has to be very clearly described what was done by the authors.

The article requires extensive formatting. The font size of the text of the tables cannot be larger that the font size of the body text. The title of the table cannot be on the different page as the table by itself. Please follow the requirements of the journal for the article preparation.

Section “Discussion” has to be dedicated to the results obtained by the authors during the presented study and refer to the outcome of the researcher rather than the other studies available in the literature. Currently this section is more introduction than discussion.   

Author Response

Reviewer #2

 The paper provides valuable experimental investigations, but I have major concerns about experimental tests, used specimens and the results which were obtained.

Authors response:

We thank the reviewer for the comments. The study was designed to fully comply with the requirements on the reproducibility of the results. Therefore, in addition to the description of the dimensions of the experimental specimen, we provide technical drawings of the specimens (Fig. 2) and technical details on analytical and numerical models. All tensile tests experiments were repeated to ensure repeatability of the results.  In case of a concern, we may provide raw data from experiments in the form of measured forces and elongations.

The issues to be considered by the Authors:

Line 72-73: The sentence “The design cell…” is incomprehensible and requires clarification.

Authors response:

We agree the original statement might be confusing. The statement was simplified by explaining the term of a unit cell.

Text Changes:

The sentence sounds in the revised manuscript: “ The unit cell has …” and the definition of the unit cell was added to the previous text: “The honeycomb structure consists of repeating unit cells of hexagonal shape.”

Line 74-75: Authors declare that the computer models were created using SolidWorks software although section 2.3 describing numerical modeling depicts FE software ABAQUS. It is not clear what for SolidWorks software was used.

Authors response:

SolidWorks was used to design the geometry of the sample while the Abaqus was used for the finite element calculations.

Text Changes:

The term computer model is replaced by the term geometry model in the revised manuscript. The version of the SolidWorks software is specified in the revised manuscript.

Line 82: It is recommended to denote the indicated hexagonal cell angle of 30 degrees in Figure 2 together with other parameters.

Authors response:

We agree.

Text Changes:

Figure 2 contains the depiction of the inner angle of the structure.

Line 82: parameter l previously was named as outer length, it is recommended to stay with the uniform terminology throughout the text.

Authors response:

We agree.

Text Changes:

Term outer diameter is replaced by the term outer length.

Line 97: statement depicting the terms of equation 3 includes Poisson's ratio although the equation does not embed this parameter. The role of Poisson's ratio has to be clarified.

Authors response:

We apologize for the missing symbols in equations caused by the inconsistent use of text processing software by the team of authors (OpenOffice and MS Word). The revised manuscript was rewritten to LaTeX to improve formatting and readability.

Text Changes:

All the equations are checked in the revised manuscript.

Section 2.3: This section describes the numerical modeling however, the description of the material models requires clarification because it is not clear what type of material model was used. Does authors use the same material models for polymer and metallic materials? The reference to the table 1which depicts elastic modulus and Poisson's ratio is not enough adequately describe the material model. Without yielding and/or strength limits it is not clear how the analysis is terminated, this has to be clarified. What is more the description of loading and boundary conditions is missing.

Authors response:

The FEA simulations were used to complement the analytical model with more realistic geometry. The analytical model assumes only linear elasticity. Therefore, the FEA model also adopts a linear elastic model defined by two material constants given in Tab. 1.  The whole loading cycle depicted in Fig. 5 was not simulated, only the initial elastic response. The loading force was chosen so, that the peak stress for all structures is below the yield stress. 

Text Changes:

Tab. 1 is modified to include the yield strength of the studied materials. Description of FEA calculation was fully rewritten concerning the study of Alwattar and Mian, 2019. A paragraph discussing boundary conditions was added in Section 2.3.

Figure 3: The title of the figure states that the depicted models in the figure have the same stress-strain response. This statement requires augmentation, graphical representation is recommended.

Authors response:

We thank the reviewer for this comment. It could be misleading to describe the response of these two materials as two independent numerical experiments. In our simulation, these models are coupled in force and deformation. The effective modulus  E* is chosen so that the equivalent material model has the same mechanical response as the honeycomb lattice structure.  The use of the equivalent elastic material model follows the methodology of Alwattar and Mian, 2019 as discussed in the previous paragraph.  As the model has the linear elastic response, we believe that the graphical representation of such a relationship would be redundant. 

Text Changes:

Description of FEA calculation was rewritten and extended to explain the use of the equivalent homogeneous model.  The study of  Alwattar and Mian, 2019 is cited in the revised manuscript where the reader can find a detailed description of the method. The caption to Fig. 3 was reformulated. 

Line 133: The statement “Polymer specimens in this study were fabricated using Stratasys J750” requires clarification. What is meant by Stratasys J750? If it’s 3D printer it has to be identified in the text.

Authors response:

Stratasys J750 is the commercial name of a 3D printer.

Text Changes:

Stated as suggested by the reviewer.

Line 135: The purpose of the support material SUP706 has to be clarified.

Authors response:

SUP706 forms the support structure during the PolyJet AM process. After that, the support material SUP 706 allows removing the support structure from the finished piece by soaking it in a cleaning solution and rinsing the printed part in water.

Text Changes:

The purpose of the support material SUP 706 is defined in section 2.4. of the revised manuscript.  

Line 141: What is meant by “cusing machine”?

Authors response:

Concept Laser metal 3D printers use the brand’s patented LaserCUSING® technology, which is based on laser sintering of metals. M2 Cusing is the brand name of the 3D printer.

Text Changes:

M2 cusing machine was changed to 3D printer M2 Cusing in the revised manuscript.

Line 158: It has to be identified what kind of equipment is entitled as “ATOS by GOM”.

Authors response:

ATOS is industrial non-contact 3D scanner using structured blue light to deliver precise scans with detailed resolution at high speed.

Text Changes:

Section 2.5 was modified by adding a description of the ATOS system by GOS and its technical parameters.

Figure 4: It has to be clarified how the results presented in the figure were obtained. Is it experimental results, numerical, or comparison of both.

Authors response:

Figure 4 presents experimental data.

Text Changes:

A statement that data were obtained from experiments were added to Fig. 4 caption.

Figure 5: The previous comment is applicable also for the Fig. 5.

Authors response:

Fig.5 presents experimental data.

Text Changes:

A statement that data were obtained from experiments were added to Fig. 5 caption.

Line 180 and Line 182: authors provide average elongation in percentage and refer to the Figure 4 although the strain in fig. 4 is not in percentage. The reference is inexpedient. Either the strain in Fig. 4 or in the text has to be changed to resemble each other.

Authors response:

We respectfully disagree. The nominal strain in dimensionless numbers follows standards for reporting stress-strain curves, e.g. ASTM E646-16.  However, the expression of elongation in percents is more readable and widely adopted in technical literature.

Line 191-192: The statement “The maximum stress is observed at the nodes” requires clarification as the nodes mentioned by the authors are not identified. It is recommended to consider the identification of the considering nodes in Fig. 6.

Authors response:

We agree that the original manuscript does not contain a definition of nodes. The text was modified as suggested by the reviewer.

Text Changes:

Section 1.1 was extended by including a definition of nodes. 

Line 193: The statement “Similar results were observed…” has to be supported by the figure.

Authors response:

We agree.

Text Changes:

Fig. 6 contains photographs of samples after tests from all three materials. 

Line 231: Authors state that the difference of the results is 1% although from the table 3 it is closer to 2%.

Authors response:

The difference between the analytical and the numerical model in the relative effective modulus is 7.36%-6.57%=0.79%.  Value 1.75% is the effective modulus of polymer material determined from the experiments. 

Text Changes:

A more accurate value is used in the revised manuscript.

Line 232-233: The statement “FEA simulation presented in Fig. 6…” has to be supplemented, the Fig. 6 has clearly depict the nodes of stress concentration.

Authors response:

We agree.

Text Changes:

The selected nodes with stress concentration are depicted in Fig. 7 in the revised manuscript. The caption of the figure was changed accordingly.

Some general comments:

From the context the experimental program is not clear. Were the tests performed by the authors or the other researches. Were the material properties tested in the lab by the authors or just the results available from the other research used. It has to be very clearly described what was done by the authors.

Authors response:

All experimental tests, measurements, and evaluations of the models were performed by the authors.

Text Changes:

All data taken from the literature are marked in the revised manuscript, e.g. citation of material data in Tab. 1. The abstract was modified to point out what was done by the authors.

The article requires extensive formatting. The font size of the text of the tables cannot be larger that the font size of the body text. The title of the table cannot be on the different page as the table by itself. Please follow the requirements of the journal for the article preparation.

Authors response:

Therefore the whole manuscript was transformed into LaTeX to improve readability and formatting using the MDPI template.

Section “Discussion” has to be dedicated to the results obtained by the authors during the presented study and refer to the outcome of the researcher rather than the other studies available in the literature. Currently this section is more introduction than discussion.

Authors response:

The discussion was rewritten as suggested by the reviewer.

Text Changes:

Part of the Discussion refereed by the reviewer was moved to the Introduction section.

Round 2

Reviewer 1 Report

In the corrected version of the manuscript, the authors have taken into account the suggestions made from their research work. The article may be considered for publication in its present form.

*Please check that equation 3 is well written. Modify the units of the effective Young’s modulus of table 3.

Author Response

Please check that equation 3 is well written.

Authors response:

We apologize for inconvenience. Eq. 3 was corrupted during the submission process. The pdf file uploaded within the review contains correct equation.

Modify the units of the effective Young’s modulus of table 3.

Authors response:

We thank for this comment. Units are shown at individual values in the revised manuscript.

Reviewer 2 Report

Dear Authors,

I have read revised version of manuscript titled: “Additive manufacturing of honeycomb lattice structure – from theoretical models to polymer and metal products”.

The paper was improved comparing to the initial version and can be published after minor revision.

The issues to be considered by the Authors:

Figure 3Check the Eq. 3, there are some layout issues which have to be corrected.

Line 103: Consider to end sentence "Two main reasons...choice." with the dot and start a new sentence afterwards, or use the small letter after the colon.

Figure 4: Check Fig. 4 there are some layout issue. The labels in the graphs embed some black parts hiding the labels itself.

Consider to use numbering (letter style as in Fig. 6) of the graphs, which will enable more convenient reference rather than referring to "left" and "right".

Author Response

Check the Eq. 3, there are some layout issues which have to be corrected.

Authors response:

We apologize for inconvenience. Eq. 3 was corrupted during the submission process. The pdf file uploaded within the review contains correct equation.

Line 103: Consider to end sentence "Two main reasons...choice." with the dot and start a new sentence afterwards, or use the small letter after the colon.

Authors response:

The manuscript was modified as suggested by the reviewer.

Figure 4: Check Fig. 4 there are some layout issue. The labels in the graphs embed some black parts hiding the labels itself.

Authors response:

Fig. 4 was corrupted during the submission process. The pdf file uploaded within the review contains correct equation.

Consider to use numbering (letter style as in Fig. 6) of the graphs, which will enable more convenient reference rather than referring to "left" and "right".

Authors response:

The manuscript was modified as suggested by the reviewer.